# Sestrin2 Expression Has Regulatory Properties and Prognostic Value in Lung Cancer

**DOI:** 10.3390/jpm10030109

**Published:** 2020-09-01

**Authors:** Hee Sung Chae, Minchan Gil, Subbroto Kumar Saha, Hee Jeung Kwak, Hwan-Woo Park, Balachandar Vellingiri, Ssang-Goo Cho

**Affiliations:** 1Department of Stem Cell and Regenerative Biotechnology, Incurable Disease Animal Model & Stem Cell Institute (IDASI), Konkuk University, 120 Neungdong-ro, Gwangjin-gu, Seoul 05029, Korea; gmltjdgk@konkuk.ac.kr (H.S.C.); minchangil@gmail.com (M.G.); subbroto@konkuk.ac.kr (S.K.S.); hjeong9581@konkuk.ac.kr (H.J.K.); 2Department of Cell Biology, Konyang University College of Medicine, Daejeon 35365, Korea; hwanwoopark@konyang.ac.kr; 3Human Molecular Cytogenetics and Stem Cell Laboratory, Department of Human Genetics and Molecular Biology, Bharathiar University, Coimbatore 641-046, India; geneticbala@buc.edu.in

**Keywords:** Sestrin2, lung cancer, knockdown, cancer progression, bioinformatics, patient survival

## Abstract

Lung cancer remains the most dangerous type of cancer despite recent progress in therapeutic modalities. Development of prognostic markers and therapeutic targets is necessary to enhance lung cancer patient survival. Sestrin family genes (Sestrin1, Sestrin2, and Sestrin3) are involved in protecting cells from stress. In particular, Sestrin2, which mainly protects cells from oxidative stress and acts as a leucine sensor protein in mammalian target of rapamycin (mTOR) signaling, is thought to affect various cancers in different ways. To investigate the role of Sestrin2 expression in lung cancer cells, we knocked down Sestrin2 in A549, a non-small cell lung cancer cell line; this resulted in reduced cell proliferation, migration, sphere formation, and drug resistance, suggesting that Sestrin2 is closely related to lung cancer progression. We analyzed Sestrin2 expression in human tissue using various bioinformatic databases and confirmed higher expression of Sestrin2 in lung cancer cells than in normal lung cells using Oncomine and the Human Protein Atlas. Moreover, analyses using Prognoscan and KMplotter showed that Sestrin2 expression is negatively correlated with the survival of lung cancer patients in multiple datasets. Co-expressed gene analysis revealed Sestrin2-regulated genes and possible associated pathways. Overall, these data suggest that Sestrin2 expression has prognostic value and that it is a possible therapeutic target in lung cancer.

## 1. Introduction

Cancer, one of the leading causes of death in modern society, poses a threat to human health worldwide. Among the various cancers, lung and bronchial cancer is the most dangerous cancer type, with 228,150 new patients and 142,670 deaths reported in 2019 in the United States alone [1]. Cancer occurrence is gradually increasing with population increase and aging, although there have been considerable advances in cancer therapy. Developments in the identification of novel cancer targets and markers are required to improve human health.

Sestrin family genes consist of Sestrin1, Sestrin2, and Sestrin3. Under conditions of stress, sestrins regulate stress-inducible metabolism and protect cells against various kinds of stressors such as hypoxia, DNA damage, and oxidative and metabolic stress [2,3]. Sestrin1, also called p53-activated gene 26 (PA26), is involved in the growth arrest and DNA damage response pathways [4]. Sestrin2, also known as hypoxia-inducible gene 95 (Hi95), is involved in mediating the response to hypoxia and is upregulated by other stressors, such as DNA damage and oxidative stress [5,6]. Sestrin3, as well as Sestrin2, is known to mediate the regulation of mammalian target of rapamycin 1 (mTORC1) and Akt activation [7,8]. Expression of these genes decreases the levels of intracellular reactive oxygen species (ROS) and promotes resistance against oxidative stress [9,10]. A recent study revealed that Sestrin1 and Sestrin2 activate Nrf2 and subsequently increase Srx, which is important for oxidative metabolism [11]. Sestrin has been shown to interact with p53 and forkhead box class O (FoxO) transcription factors and mediate antioxidant regulation [12]. Activation of 5′ adenosine monophosphate-activated protein kinase (AMPK) and inhibition of mTORC1 are important for cell cycle and cell lifespan [13]. Since sestrins can modulate pathways of cellular metabolism, sestrin expression seems to play an important role in prolonging life and inhibiting aging [2].

Sestrin2 is an intracellular leucine sensor protein that negatively regulates mTORC1 signaling by binding to GAP Activity Towards Rags 2 (GATOR2), a subunit of the GATOR complex, in the absence of leucine. In the presence of leucine, Sestrin2 detaches from GATOR2 and consequently activates mTORC1 [14,15,16]. Sestrin2 plays a variety of roles throughout the body and is responsible for mediating the cellular response against various environmental stressors [2]. Genotoxic stresses, such as UV or gamma irradiation, and genotoxic molecules promote the transcription of Sestrin1 and Sestrin2 through the p53 pathway, resulting in cell cycle inhibition and modulation of metabolism in the stressed cells [17]. Oxidative stress activates the Nuclear factor erythroid-2-related factor 2 (NRF2) and Jun N-terminal kinase-Activator protein 1 (JNK-AP1) pathways, which induce the expression of Sestrin2 [18,19]. Sestrin2 has also been shown to act as a tumor suppressor gene in various cancers [20,21,22]. In colorectal cancer, Sestrin2 suppresses mTORC1 signaling by activating AMPK/mTORC pathway, resulting in the suppression of tumor cell growth [21]. Sestrin2 knockdown accelerates colorectal carcinogenesis [22]. Moreover, Sestrin2 is known to be downregulated in bladder cancer, and when Sestrin2 is induced in response to mitogen-activated protein kinase 8 (MAPK8)-JUN)-dependent transcription, it suppresses bladder cancer growth [23]. However, contrary to these results, Sestrin2 is still expressed in various cancers and may be necessary to increase cancer viability under certain conditions [24].

Based on previous reports, the present study aims to investigate the role of Sestrin2 in the survival, migration, and sphere formation of lung cancer cells. Further, this study also aims to conduct bioinformatic analyses for gene expression, prognostic value, and potential related pathways in human samples using various cancer gene expression databases. The outcome of this study with respect to Sestrin2 may indicate its potential role in prognostics and therapeutics for lung cancer.

## 2. Materials and Methods

### 2.1. Cell Culture

Human lung carcinoma cell line A549 was purchased from The Korea Cell Line Bank (Seoul, Korea). A549 was cultured in RPMI 1640 medium (Sigma-Aldrich, St. Louis, MO, USA) supplemented with 10% heat-inactivated fetal bovine serum (FBS, Gibco, Thermo Fisher Scientific Ltd., Waltham, MA, USA), 100 U/mL penicillin, and 100 mg/mL streptomycin (Gibco). Cells were seeded in cell culture plates (SPL Lifesciences, Pocheon-si, Korea) and maintained at 37 °C in a humidified atmosphere containing 5% CO_2_.

### 2.2. Lentivirus Production and Infection

The short hairpin (shRNA) lentiviral plasmids for Sestrin2 knockdown (shSESN2-1 and shSESN2-2) and scramble control were purchased from VectorBuilder (Chicago, IL, USA). The sequence of shRNA was designed as previously reported [25,26]. pLSLPw-shLUC and shSESN2 were provided by Dr. Andrei Budanov [25]. To prepare the lentivirus, we cultured human embryonic kidney (HEK) 293T cells up to 80% confluence in 6-well plates and transfected with scramble and Sestrin2-targeted shRNA plasmids using lipofectamine 3000 (Thermo Fisher Scientific Ltd., Waltham, MA, USA). After 48 h, the virus-containing medium was collected and filtered with a 0.45 µm membrane filter. The scramble and shRNA targeting SESN2 (shSESN2) lentivirus supernatant were used to infect A549 cells, which were incubated overnight. Afterward, the media was replaced with fresh culture media and the cells were grown. Puromycin was used for treatment 24 h after media change, and the concentration was 4 μg/mL.

### 2.3. Total RNA Extraction and RT-PCR

Total RNA was extracted using Labozol (Labopass, Cosmogenetech, Seoul, Korea) with the experimental protocol provided. The concentration of total RNA was measured using Nanodrop (IMPLEN, CA, USA). Complementary DNA (cDNA) was synthesized using 2 µg of total RNA with Moloney Murine Leukemia Virus (MMLV) reverse transcriptase (Promega) as per the experimental protocol provided. RT-PCR was performed using r-Tap Plus Master Mix (Elpis Biotech, Daejeon, Korea), and PCR products were analyzed by electrophoresis on a ~1.5% agarose gel containing ethidium bromide (EtBr) and bands were observed under UV light. Relative expression was measured using ImageJ (https://imagej.net/). Primer sequences are listed in Appendix A.

### 2.4. Cell Survival Assay

For cell proliferation analysis, control and Sestrin2 knockdown cells were seeded in 6-well plates (5 × 10^4^ cells/well) and cultured for 24, 48, 72, and 96 h. Cells were counted using a hemocytometer following trypan blue staining. Cell proliferation assay was also carried out using EZ-cytox reagent (DoGen, Seoul, Korea). Around 1 × 10^4^ cells/well were seeded in 96-well plate and cultured. EZ-cytox was added at a ratio of 1 to 10 and held in an incubator for 2 h. Afterward, the absorbance was measured at 450 nm using a fluorescence microplate reader. To observe the drug sensitivity, the cells were seeded in 6-well plates (1 × 10^5^ cells/well) and grown. After 24 h, doxorubicin and cisplatin were added to each well and mixed well. The final concentration was 1 μM for doxorubicin and 10 μM for cisplatin. After 24 h of treatments, the cells were counted using hemocytometer following trypan blue staining.

### 2.5. Wound Healing Assay

For the wound healing assay, 95% confluent cells were cultured in a 6-well plate. Cells were treated with mitomycin C (10 µg/mL) for 2 h, then the monolayer was scratched using a 200 µL tip and the media was replaced with fresh culture media with 10% fetal bovine serum (FBS). The wound area was marked, and photos were taken every 12 h. Pictures were analyzed, and closure percentage was measured using ImageJ.

### 2.6. Sphere-Forming Assay

Cells (1 × 10^5^) were seeded in a non-coated 60 mm petri dish (SPL Lifesciences, catalogue number 11035) and cultured in the presence of sphere-forming media (serum-free DMEM/F12 media containing B27 supplement, 20 ng/mL epidermal growth factor (EGF) (Sigma-Aldrich, St. Louis, MO, USA), 10 µg/mL insulin (Sigma-Aldrich), and 1% bovine serum albumin (Sigma-Aldrich)). After 5 days, spheres were collected and stained with crystal violet (Sigma-Aldrich). Sphere sizes were measured using ImageJ.

### 2.7. Dichlorodihydrofluorescein Diacetate (DCFDA) Cellular ROS Assay

Cells (1 × 10^4^) were seeded in 96-well plates. After 24 h, 2′,7′-dichlorodihydrofluorescein diacetate (DCFDA) (Invitrogen, Waltham, MA, USA) was added to the cells at a final concentration of 10 mM, followed by incubation at 37 °C for 30 min. After removing the DCFDA media, cells were washed with DPBS and fluorescence was measured immediately on a fluorescence microplate reader (SpectraMAX, Molecular Devices). For flow cytometry, cells (5 × 10^5^) were seeded and cultured in 6-well plates. After 24 h, DCFDA was added to cells and incubated for 30 min. DCFDA media was removed, and cells were washed with Dulbecco’s Phosphate-Buffered Saline (DPBS), detached by trypsin, and analyzed by flow cytometry (FACSCalibur, Becton Dickinson).

### 2.8. Oncomine Database Analysis

The expression of Sestrin2 mRNA was analyzed using Oncomine with the Okayama Lung Statistics and Selamat Lung Statistics datasets (https://www.oncomine.org/resource/login.html) [27,28]. mRNA expression was compared between lung cancer and normal tissues using parameters with a threshold *p*-value of 1 × 10^−4^ and gene ranking in the top 10%. We also analyzed genes co-expressed with Sestrin2 in the Bass lung dataset.

### 2.9. The Human Protein Atlas

Expression of Sestrin2 protein in lung cancer tissue and normal tissue was analyzed using the Human Protein Atlas (https://www.proteinatlas.org/). The normal lung tissue from Patient 2268 was compared with lung cancer tissue from patient 3391 that stained with Sestrin2 antibody (HPA018191, Sigma-Aldrich, St. Louis, MO, USA).

### 2.10. Prognoscan and Kaplan-Meier Plotter

The correlation between Sestrin2 expression and survival rate in lung cancer patients was analyzed using Prognoscan and the Kaplan-Meier plotter database. Prognoscan is a database that includes prognostic data for various cancers (http://dna00.bio.kyutech.ac.jp/PrognoScan/) [29]. Gene Expression Omnibus (GSE)3141–overall survival (hazard ratio = 2.38) and GSE11117–overall survival (hazard ratio = 2.59) were analyzed using Prognoscan with a Cox *p*-value < 0.05. The Kaplan-Meier plotter database was used to analyze mRNA Affymetrix Genechip and RNA-sequencing datasets for lung squamous cell carcinoma patient.

### 2.11. cBioPortal Database Analysis

The mutation and alteration of Sestrin2 were analyzed using cBioportal, a free-access bioinformatic website (http://www.cbioportal.org/) [30]. cBioportal provides clinical characteristic data from 225 cancer studies in The Cancer Genome Atlas (TCGA) datasets. The mutations in 2704 cases of lung cancer and gene alteration in 2324 cases were analyzed. Copy number alteration analysis was performed using the GISTIC (Genomic Identification of Significant Targets in Cancer) algorithm and plotted using TCGA mRNA expression data.

### 2.12. Enrichr Gene Ontology (GO) Analysis

To analyze the ontology of Sestrin2 and co-expressed genes, the Enrichr database was used (https://amp.pharm.mssm.edu/Enrichr) [31]. GO and pathway analyses were visualized as a bar graph. GO biological process, molecular function, cellular component, and Kyoto Encyclopedia of Genes and Genomes information were also included in the analysis.

### 2.13. Statistical Analysis

Data were analyzed using GraphPad Prism 6 (Sandiego, CA, USA) and Excel 2006 (Microsoft Corporation, Redmond, WA, USA). Statistical analyses were conducted using a *T*-test and statistical significance was defined as * *p* < 0.05.

## 3. Results

### 3.1. Knockdown of Sestrin2 in a Lung Cancer Cell Line Leads to Reduced Cancer Cell Survival and Migration

We detected relatively high Sestrin2 expression in A549, a non-small cell lung cancer cell line compared to other cell lines tested (Appendix A). To investigate the effect of Sestrin2 on lung cancer cells, we examined the effects of Sestrin2 knockdown in these cells. Knockdown was performed using Sestrin2-targeted shRNA cloned in a lentiviral vector. Reverse transcription-polymerase chain reaction (RT-PCR) analysis revealed that expression of Sestrin2 was reduced by shRNA in A549 cells (Figure 1A). Sestrin2 expression was decreased 72% by shSESN2-1 and 92% by shSESN2-2 compared to the scramble control. To observe the effect of Sestrin2 in cancer cells, we compared the viability of A549 cells treated with both shSESN2 and scramble control. The number of Sestrin2 knockdown cells with shSESN2-1 and SESN2-2 was significantly reduced compared to that in the scramble control (Figure 1B and Appendix A). We performed a wound healing assay with A549 cells to examine the effect of Sestrin2 expression on cancer cell migration (Figure 1C). The results showed that the gap distance of the wound in scramble control cells was more closed than that in either Sestrin2 knockdown cultures. The expression of epithelial–mesenchymal transition (EMT) markers, which might contribute to cancer metastasis, was also observed (Figure 1D). RT-PCR revealed that the expression of EMT markers (Vimentin, Snail, *ZEB1*) was significantly reduced in Sestrin2 knockdown cells compared to that in scramble cells. Overall, we suggest that Sestrin2 expression is related to survival and migration in the A549 lung cancer cell line.

### 3.2. Knockdown of Sestrin2 in Lung Cancer Cells Decreases Cancer Cell Stemness and Drug Resistance

To investigate the role of Sestrin2 in cancer cell stemness, we determined the expression of stemness marker genes by RT-PCR (Figure 2A). Expression of stemness markers Oct4, Sox2, and Nanog was decreased in Sestrin2-knockdown A549 cells compared to that in the scramble control. The effect of Sestrin2 gene on cancer stemness by sphere-forming assay was also determined (Figure 2B). The size of the spheres formed by the Sestrin2 knockdown A549 cells was smaller than that formed by scramble A549 cells. This result showed that Sestrin2 knockdown reduced lung cancer stemness. To evaluate the effect of Sestrin2 on drug sensitivity, the expression of drug resistance marker genes (*ABCG*, *ABCA2*) was determined using RT-PCR (Figure 2C). Expression of the drug resistance marker genes was decreased in Sestrin2 knockdown A549 cells compared to that in cells with scramble (Figure 2C). In addition, cell survival assay was performed on knockdown and scramble cells treated with the anticancer drugs doxorubicin and cisplatin, which induce oxidative stress by increasing the ROS level [32] (Figure 2D). The survival rate of Sestrin2 knockdown cells was significantly decreased compared to that of scramble cells regardless of anticancer drug treatment. However, the percentage of cells recovered with doxorubicin treatment was 49.7% for scramble control, 41.6% for shSESN2-1, 41.1% for shSESN2-2 compared to no treatment control. The reduced recovery rate of Sestrin2 knockdown cells in doxorubicin treatment suggests that cells became more sensitive to doxorubicin treatment. These results suggest that the expression of Sestrin2 could be involved in mediating the development of cancer stemness and drug resistance in lung cancer cell lines.

### 3.3. Expression of Sestrin2 is Related to ROS Regulation in A549 Lung Cancer Cells

NF-E2-related factor 2 (*NRF2*) is a critical transcription factor regulating intracellular antioxidants and detoxification enzymes [33]. In cancers, the NRF2-mediated antioxidant pathways protect cells from drugs such as doxorubicin and cisplatin [34]. Because Sestrin2 activates the *NRF2* pathway in cancer cells [11], the effect of Sestrin2 knockdown on *NRF2* and oxidative status of A549 cells was investigated. For ROS measurement by DCFDA assay, Sestrin2 knockdown cells without GFP expression were generated, and the knockdown of Sestrin2 and downregulation of *NRF2* and heme oxygenase (*HO-1*) were confirmed in A549 cells (Figure 3A). Reduced expression of *NRF2* and *HO-1* were also observed in Sestrin2 knockdown A549 cells with the shRNA vectors used in Figure 1 and Figure 2 (Appendix A). The intracellular ROS level was then measured using the DCFDA assay. In the Sestrin2 knockdown cells, ROS levels were significantly increased by nearly threefold (Figure 3B). The increase in ROS levels was also indicated by flow cytometry (Figure 3C). These results suggest that Sestrin2 affects the regulation of the NRF2-HO-1 pathway and ROS level in A549 cancer cells.

### 3.4. Sestrin2 Expression and Correlation with Patient Survival in Lung Cancer

Knockdown of Sestrin2 in A549 cells suppressed cancer cell properties such as proliferation, migration, stemness, and drug resistance, which are critical to cancer progression. To examine the role of Sestrin2 in human lung cancer, we used a publicly available gene expression database of cancer tissues. In analysis using the Oncomine database, Sestrin2 mRNA expression was higher in lung cancer tissue than in normal lung tissue in the Okayama Lung Statistics dataset (fold change, 1.295; *p*-value: 3.34 × 10^−7^) (Figure 4A). Histochemistry data for lung cancer using the Human Protein Atlas revealed that Sestrin2 protein was highly expressed in lung cancer (Figure 4B). The lung tumor was strongly stained by Sestrin2 antibody HPA018191 (patient ID = 3391), while pneumocytes of the normal lung were stained less strongly (patient ID = 2268). Based on these results, we suggest Sestrin2 is highly expressed in lung cancer than in the normal tissue.

The correlation between Sestrin2 expression and lung cancer patient survival was analyzed using KM plotter and Prognoscan. In meta-analysis using Prognoscan database, patient overall survival was significantly correlated with Sestrin2 expression in three lung cancer datasets (Figure 4C). The survival rate of the lung cancer patient group with high Sestrin2 expression was lower than that in the patient group with low Sestrin2 expression in the GSE3141 dataset (*p*-value: 0.0037) and in the GSE11117 (*p*-value: 0.023) (Figure 4D). However, Sestrin2 expression is positively correlated with patient overall survival in the GSE13213. In KM plotter analysis, the lung squamous carcinoma patient group with higher Sestrin2 expression had worse overall survival than the patient group with lower Sestrin2 expression in the RNAseq dataset (*p*-value: 0.042) and in the Affymetrix Genechip dataset with probe 223195_s_at (*p*-value: 0.023) (Figure 4D). Overall, Sestrin2 expression is negatively correlated with survival of patients with lung cancer in multiple lung cancer expression datasets.

### 3.5. Mutation and Alteration of Sestrin2 Gene in Lung Cancer

Mutations in lung cancer patients were analyzed using cBioportal web. The mutations of Sestrin2 in lung cancer patients were analyzed across 4510 samples from 4154 patients in 16 studies. Thirty-six mutations were analyzed in the Sestrin2 protein (Figure 5A). Mutation occurred in 2.19% of the samples, and deep deletion occurred in 0.55% of the samples, resulting in gene alteration in 2.73% of the lung adenocarcinoma Broad dataset (Figure 5B,C). Expression of Sestrin2 was also analyzed based on gene alteration (Figure 5D). Expression of Sestrin2 increased the following in order: deep deletion (DD), shallow deletion (SD), diploid (D), gain (G), and amplification (A) in lung adenocarcinoma and lung squamous cell carcinoma. We suggest that mRNA expression of Sestrin2 is associated with copy number alteration.

### 3.6. Genes co-Expressed with Sestrin2 in Lung Cancer

To explore Sestrin2-related pathways, the genes co-expressed with Sestrin2 were analyzed in the Bass lung dataset using Oncomine (Figure 6A). Highly co-expressed genes are listed by correlation rate. Ontology analysis was performed with Sestrin2 and its 11 co-expressed genes using Enrichr (Figure 6B). In GO analysis, Sestrin2 and its positively co-expressed genes were analyzed in biological processes, molecular function, and cellular components. In GO biological process, regulation of cell cycle process and protein tetramerization were highly related. Moreover, DNA-directed RNA polymerase 2 holoenzyme and RNA polymerase II transcription factor complex were associated in GO cellular component. In GO molecular function analysis, the most highly ranked terms were N-6 methyladenosine-containing RNA binding and leucine binding. In the Kyoto Encyclopedia of Genes and Genomes (KEGG) pathway database, the term ‘basal transcription factors’ was related to Sestrin2 and positively co-expressed genes. To check whether Sestrin2 expression regulates the expression of co-expressed genes, we analyzed the expression of the top 5 genes in Sestrin2-knockdown A549 cells by RT-PCR. Expression of all top 5 co-expressed genes was reduced in Sestrin2-knockdown cells (Figure 6C). These co-expressed gene profiles implied that Sestrin2 expression could regulate the expression of multiple co-expressed genes, possibly related to tumor progression.

## 4. Discussion

Sestrin2, a highly evolutionarily conserved protein, is involved in mediating cellular responses to various stressors. It has a protective effect against physiological and pathological conditions, mainly through regulating oxidative stress and inflammation [35]. Sestrin2 is a leucine sensor protein that regulates mTORC1 signaling that is related to cell proliferation and growth. Sestrin2 also plays an important role in cell protection and homeostasis, mainly by the downregulation of ROS and mTOR signaling [36]. It belongs to the family of stress-inducible proteins that has a pivotal role in regulating antioxidants, autophagy, and apoptosis, thereby enabling protection from any form of DNA damage, oxidative stress, hypoxia, or metabolic stress [37]. As a result, Sestrin2 keeps cells healthy, and it has been suggested that it may prevent cancer. In this paper, we investigated the effect of Sestrin2 knockdown in A549, a non-small cell lung cancer cell line, and analyzed the prognostic value of Sestrin2 expression in human lung cancer by employing various bioinformatic tools on various lung cancer datasets.

In the A549 lung cancer cell line, Sestrin2 knockdown led to downregulation of cancer properties, confirming the oncogenic function of Sestrin2. Our data showed that Sestrin2 knockdown resulted in reduced tumor cell proliferation and migration (Figure 1). Moreover, in the sphere-forming assay, the size of sphere was significantly decreased upon Sestrin2 knockdown in A549 cells (Figure 2). Based on these results, we suggest that Sestrin2 has oncogenic effects in lung cancer cells. Consistently, lung cancer cells from Sestrin2-deficient mice showed slower growth rates than did those from wild type mice [38]. However, Sestrin2 knockdown has also been reported to promote proliferation of cancer cells, inhibit apoptosis of cells [38,39,40], and enhance migration in the wound healing assay [40], which is in complete contrast to our results. Several previous studies have reported that Sestrin2 can work as a tumor suppressor gene in various cancers [20,21,22]. Sestrin2 was proposed to regulate AMPK/mTORC pathway activation and tumor cell growth in colorectal cancer [21], and Sestrin2 knockdown accelerated colorectal carcinogenesis [22]. Sestrin2 is also known to be downregulated in bladder cancer, and Sestrin2 expression upon MAPK-JUN-dependent transcription leads to the suppression of bladder cancer growth [23]. However, contrary to these results, Sestrin2 is still expressed in various cancers and may be necessary to increase cancer viability under certain conditions [24]. In lung cancers, there have also been conflicting reports about the role of Sestrin2. Downregulated Sestrin2 expression reduces death-receptor-induced apoptosis in lung cancer cell lines [25]. Sestrin2 expression is positively correlated with patient survival in 210 non-small cell lung cancer (NSCLC) tissue samples [41]. However, in glutamine-depleted lung cancer cells, upregulated Sestrin2 increases cell survival [42]. Based on contradictory reports, Sestrin2 expression studies on cancer progression could reveal opposite results, likely dependent on different cellular conditions, which need to be characterized in detail in future studies. Differences in the effects of Sestrin2 knockdown on cellular proliferation and migration might be due to differences in culture conditions between laboratories, originating from use of different reagents such as batches of fetal bovine serum, or different protocol details such as cell numbers used for each assay or subculture.

Sestrin2 knockdown increased the intracellular ROS concentration in A549 cells with reduced expression of antioxidant genes nrf-2 and HO-1 (Figure 3). In A549 cells, reduction in intracellular ROS concentration by the antioxidant molecule N-acetyl cysteine enhanced cellular proliferation [43], suggesting that reduced intracellular ROS level is a favorable condition for proliferation. Therefore, increased amount of ROS in Sestrin2 knockdown A549 cells may be a negative regulator of cellular proliferation and/or apoptosis induction. Treatment with doxorubicin or cisplatin induces cell death via the increase of ROS in A549 cells [44,45]. Sensitization to doxorubicin and cisplatin in Sestrin2 knockdown cell was not apparently detectable because impaired proliferation of Sestrin2 knockdown cells already reduced the recovered number of cells in no treatment control cells. However, the reduction in the rate of survival in Sestrin2 knockdown A549 cells in the doxorubicin-treated group is larger than that of no treatment control, suggesting sensitization to doxorubicin treatment in Sestrin2 knockdown cells (Figure 2D). Overall, our in vitro Sestrin2 knockdown experiment supports the assumption that reduced expression of Sestrin2 could be a favorable prognostic marker for survival of lung cancer patients.

In addition, we analyzed gene expression databases with various web tools to investigate the expression and prognostic value of Sestrin2 in lung cancers. In a dataset in the Oncomine database, mRNA expression of Sestrin2 was upregulated in lung cancer compared to that in normal tissue. Sestrin2 protein expression was upregulated in lung cancer patients in the Human Protein Atlas. In addition, Sestrin2 mRNA expression was negatively correlated with the survival of lung cancer patients in multiple datasets. These results suggest that overexpressed Sestrin2 could have a poor prognostic value in lung cancer, which was in agreement with our in vitro data using A549 cells.

In the tumorigenesis processes, somatic loss-of-function or gain-of-function alterations in specific genes could have carcinogenic effects. However, mutations in the Sestrin2 gene have not been studied. Therefore, we used cBioPortal to determine mutations and CNAs in Sestrin2 gene. we a found several missense and truncating mutations within Sestrin2 protein-coding sequences (Figure 5A). The impact of each mutation in Sestrin2 has not been experimentally validated. We also found that expression of Sestrin2 was associated with the copy number alterations. This result implies that augmented Sestrin2 expression could be caused by the copy number alteration in lung cancer cells.

The co-expressed gene profile of Sestrin2 revealed pathways associated with Sestrin2 (Figure 6). The most highly rated gene ontology terms of GO biological process was regulation of cell cycle, which is closely related to cancer growth. Highly ranked terms in GO cellular component included RNA polymerase II (GO:0016591 and GO:0090575). Other terms including core mediator complex, STAGA complex, SAGA complex, transcription factor TFTC complex, condensation nuclear chromosome, and condensed chromosome are related to histone acetylation and chromosomal condensation. Most of the terms for GO cellular component suggested that Sestrin2 may be involved in transcriptional control through chromosomal condensation. The most highly ranked term in GO molecular function, N6-methyladenosine-containing RNA binding, may also be involved in transcriptional control; N6-methyladenosine is the most frequent mRNA modification significantly affecting gene expression and splicing [46]. KEGG pathway analysis includes p53 and mTOR signaling pathways, which were already known from previous studies [16]. Most importantly, knockdown of Sestrin2 also suppressed the expression of most highly correlated genes, which means that Sestrin2 is the upstream regulator of these associated pathways. This co-expressed gene analysis strongly suggests that Sestrin2 may be a key regulator of gene expression in lung cancer cells, which remains to be elucidated in further studies.

In this study, we investigated the impact of Sestrin2 expression in lung cancer with knockdown in a lung cancer cell line in vitro, and bioinformatic analysis using gene expression datasets of lung cancer. Further subsequent investigation using lung cancer cells including key cancer pathway analysis and in vivo study using animal model remains to be studied to elucidate the underlying mechanism of Sestrin2 in lung cancer.

## 5. Conclusions

In conclusion, Sestrin2 knockdown in lung cancer cells suppressed cancer cell properties, including proliferation, migration, stemness, and drug resistance. In human cancer expression datasets, increased expression of Sestrin2 and correlation of Sestrin2 expression with lung cancer patient survival was observed. Sestrin2 may be an upstream regulatory gene for its associated pathways. Thus, Sestrin2 may have prognostic value and serve as a therapeutic target in lung cancer.

## Figures and Tables

**Figure 1 jpm-10-00109-f001:**
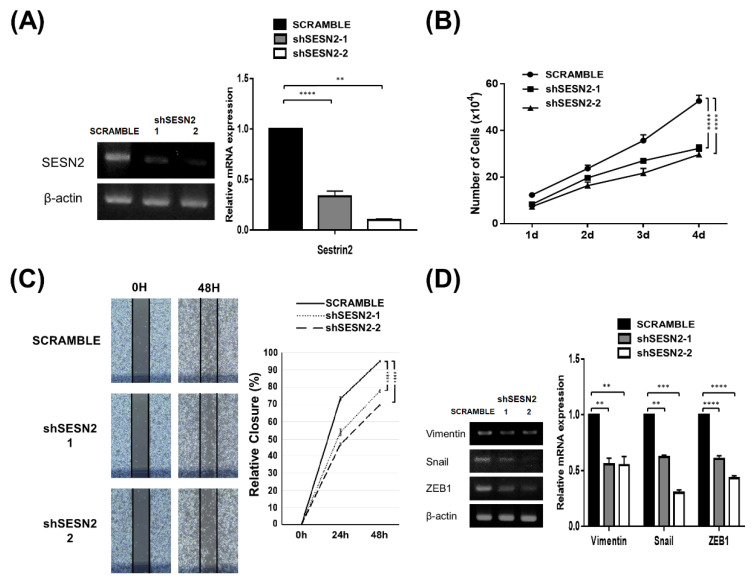
Survival and migration were decreased in response to Sestrin2 knockdown in A549 lung cancer cells. (**A**) Expression of Sestrin2 in shSESN2-1, shSESN2-2, and scramble A549 cells as measured by RT-PCR followed by subsequent agarose gel analysis. Band density was measured with ImageJ and is plotted as the value relative to scramble control. (**B**) Survival of scramble, shSESN2-1, and shSESN2-2 cells was measured using trypan blue staining. The number of cells was counted using a hemocytometer. (**C**) The wound healing assay revealed cell movement capacity. Cells were observed at the indicated time, and closure percentage is plotted. The photo was taken under an inverted light microscope and closure percentage was measured using ImageJ. (**D**) The mRNA expression of epithelial–mesenchymal transition markers (Vimentin, Snail, ZEB1) was downregulated in Sestrin2 knockdown cells compared to that in scramble cells. Band density was measured using ImageJ and is plotted on the right (** *p* < 0.01; *** *p* < 0.005; **** *p* < 0.0001).

**Figure 2 jpm-10-00109-f002:**
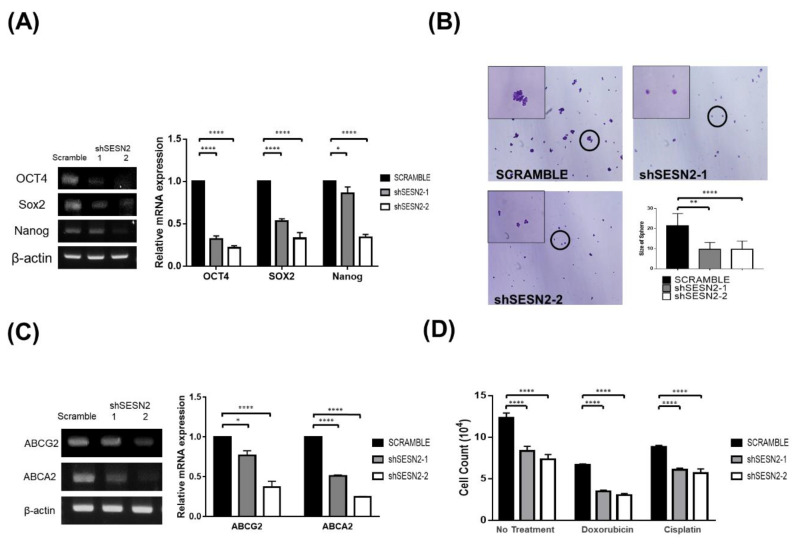
Effect of Sestrin2 expression on stemness and drug resistance in A549 cells. (**A**) Expression of stemness marker in scramble and shSESN2 A549 cells was analyzed using RT-PCR. mRNA expression relative to scramble control is shown in the graph. (**B**) Sphere-forming assay of scramble and Sestrin2 knockdown A549 cells. Cells were seeded in a petri dish and cultured in sphere-forming media. Spheres were evaluated after 5 days of culture using crystal violet, and then photos were taken. The size of the spheres was measured by ImageJ. The enlarged photo at the top left represents spheres in a circle. (**C**) Expression of the drug resistance marker genes (*ABCG2*, *ABCA2*) in scramble and shSESN2 A549 cells was analyzed using RT-PCR. (**D**) Drug resistance assay with doxorubicin and cisplatin, ROS-generating anticancer drugs. Scrambled control, shSESN2-1, and shSESN2-2 cells were treated with 1 μM doxorubicin or 10 μM cisplatin for 24 h and subjected to cell counting with trypan exclusion (* *p* < 0.05; ** *p* < 0.01; **** *p* < 0.0001).

**Figure 3 jpm-10-00109-f003:**
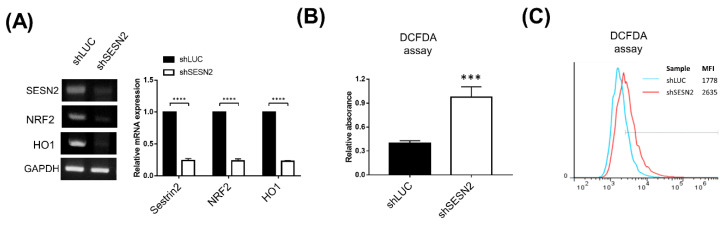
Sestrin2 knockdown leads to reactive oxygen species (ROS) overproduction by inhibiting the oxidative stress response. (**A**) Expression of *NRF2* and *HO-1* in control and shSESN2 A549 cells measured by RT-PCR. A549 cells were transduced with lentiviral pLSLPw-shLUC and shSESN2 plasmids. (**B**) 2′,7′-Dichlorodihydrofluorescein diacetate (DCFDA) cellular ROS assay. A549 cells were stained with DCFDA and washed. The emitted fluorescence was measured using a fluorescence microplate reader (**B**) and flow cytometer (**C**). MFI: mean fluorescence intensity (*** *p* < 0.005; **** *p* < 0.0001).

**Figure 4 jpm-10-00109-f004:**
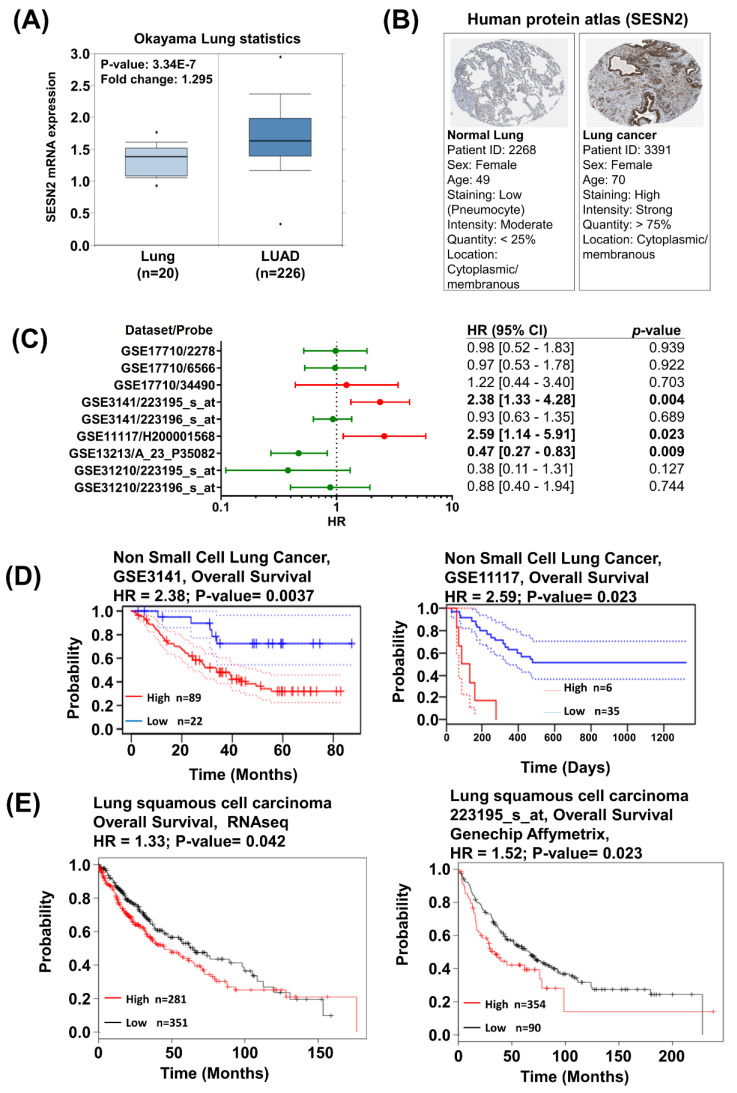
Sestrin2 mRNA expression analysis in lung cancer patients using various bioinformatic databases. (**A**) Oncomine database analysis of Okayama Lung statistics comparing Sestrin2 mRNA expression in normal lung with that in lung cancer. (**B**) Human Protein Atlas analysis for patient tissue staining. The normal lung tissue is stained with a low amount of Sestrin2 antibody (patient ID = 2268) while the lung cancer tissue is stained with a high amount of Sestrin2 antibody (patient ID = 3391) (**C**) Forest plots of GEO datasets evaluating association of sestrin2 expression and overall survival in lung cancer datasets in Prognoscan database. Hazard ratio (HR) with 95% confidential interval (CI) and p-values are labeled in the right column of each forest plot. (**D**) The survival rate graph compares high (red) and low (blue) Sestrin2 expression in non-small cell lung cancer patients. Prognoscan database analysis survival curve plotter using GSE3141–overall survival (hazard ratio = 2.38, *p*-value = 0.0037) (high *n* = 89, low *n* = 22) and GSE11117–overall survival (hazard ratio = 2.59, *p*-value = 0.023) (high *n* = 6, low *n* = 35) datasets. (**E**) The survival rate graph compares high (red) and low (black) Sestrin2 expression in lung squamous cell carcinoma patients. The 223195_s_at dataset was analyzed with RNAseq and Affymetrix Genechip using KM plotter.

**Figure 5 jpm-10-00109-f005:**
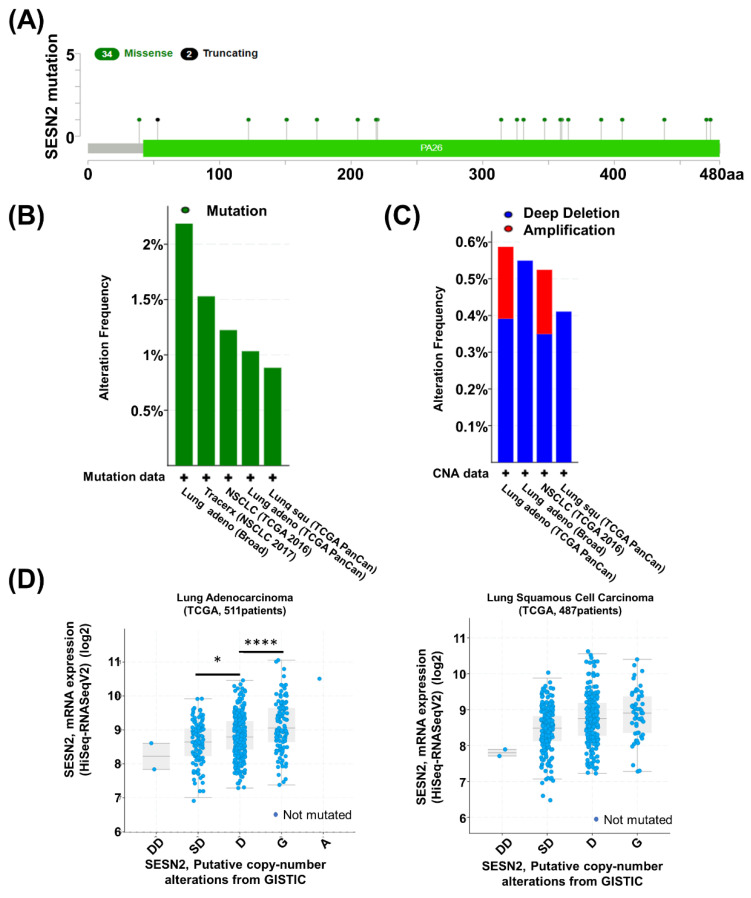
Sestrin2 mutation and alteration in TCGA lung cancer. (**A**) The mutation plot shows the location and type of mutation in Sestrin2. (**B**) Sestrin2 mutation analysis using cBioportal. Green: mutation. (**C**) Sestrin2 alteration analysis using cBioportal. Red: amplification; blue: deep deletion. (**D**) Copy number alteration of Sestrin2 mRNA expression in TCGA lung adenocarcinoma and TCGA lung squamous cell cancer datasets. Sestrin2 expression positively related to the copy number alteration status, deep deletion (DD), shadow deletion (SD), diploid (D), gain (G), and amplification (A). (* *p* < 0.05; **** *p* < 0.0001).

**Figure 6 jpm-10-00109-f006:**
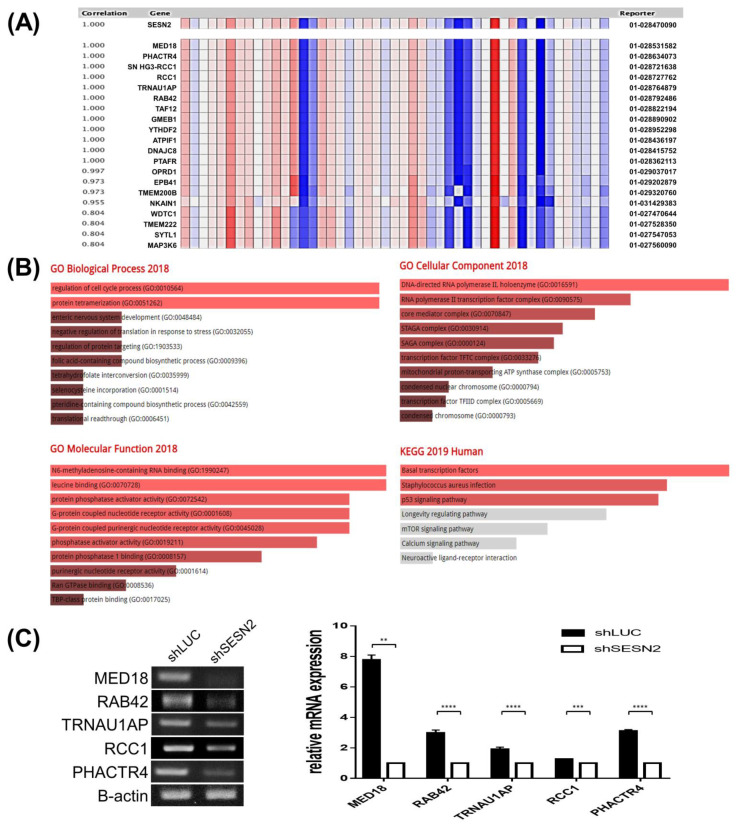
Sestrin2 co-expression gene analysis using RT-PCR (**A**) Co-expression gene analysis in the Bass lung dataset using the Oncomine database. (**B**) GO and KEGG analysis with Sestrin2 and co-expressed genes using Enrichr; bar graph listed by p-value. The brighter the bar color, the more significant the related pathway. (**C**) The expression of top 5 co-expression genes was downregulated in Sestrin2 knockdown A549 cell as shown by RT-PCR. Fold-change was measured by ImageJ; graph shown to the right. (** *p* < 0.01; *** *p* < 0.005; **** *p* < 0.0001).

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
