# Peer review of "Sestrin2 Expression Has Regulatory Properties and Prognostic Value in Lung Cancer"

_jpm, 2020, doi:10.3390/jpm10030109_

Round 1

Reviewer 1 Report

Chae HS et al. wrote a paper entitled Sestrin2 expression has regulatory properties and prognostic value in lung cancer describing oncogenic potential of Sestrin2. In the paper the group used biological and bioinformatical approach to tackle the role of Sestrin2 in proliferation, migration, stemness, and drug resistance of lung cancer cells.  

Major points:

  • In figure 2d there is no non-treated control for each cell line model. The effect showed in this figure could be also a result of basal survival impairment with shSESN2 proved by the group in figure 1. Therefore, in order to be able to state that Sestrin2 knockdown is sensitizing cells toward doxorubicin and cisplatin treatments it is necessary to have proper controls.
  • To further prove that Sestrin2 knockdown is sensitizing A549 cells to doxorubicin and cisplatin treatment through ROS it would be necessary to measure ROS levels with aforementioned treatments and try to rescue the observed effect with ROS scavenger agents (such as N-acetyl-cysteine).
  • Re-expressing Sestrin2 in knockdown cells should be able to rescue levels of EMT (figure 1d), stemness (figure 2a), and drug resistance (figure 2c) markers, as well as NRF2 and HO1 levels and enhance the statement that Sestrin2 is directly related to mentioned pathways.
  • The showed experiments are mostly done on RNA level. It would increase the quality of generated data if the same regulations were proven on the protein level.

technical issues:

  • cell survival assay using hemocytometer is quite insensitive and time-consuming, authors can use MTT or SRB assay instead.
  • Wound healing assay, authors need to clarify the condition if it is FCS dependent or not (a migration or a proliferation)
  • i am wondering which petri dish was used for this suspension culture.
  • Bioinformatic analysis using published data is important for this manuscript? Authors can add more experimental data instead of bioinformatics, like cancer stem cell surface marker analysis, in vitro dilution assay, colony formation assay, if possible measure the tumorigenesis capacity of sestrin+/- cells in animal model

Minor points:

    • Lines 41-43 and lines 50-53 are both referring to Sestrins’ antioxidant activity and could be brought together.
    • Lines 45-50: When describing all three Sestrin proteins, Sestrin3 was introduced as mTORC1 regulator, while Sestrin2 is actually the most investigated Sestrin that is known to regulate mTORC1 (as later mentioned in the line 57).
    • Lines 51-53: do not clearly indicate which Sestrin in particular they refer to.
    • Lines 65-68: activation of AMPK/mTORC pathway is resulting in reduction of mTORC1 activity. Hence, the sentence in aforementioned lines could be rephrased. Furthermore, statement that Sestrin2 is suppressing mTORC1 e
    • xpression is not fully accurate. Sestrin2 downregulates mTORC1 activity but, to my knowledge, it is still not proven that it suppresses its expression.
    • Line 168: It is stated that the group detected high expression of Sestrin2 in A549 cells, but there is no data to prove that statement, or citation to refer it to.
    • In figure 1b y-axis is labeled as relative cell count and, in the description, it is indicated that the figure is showing survival. Is y-axis in %?
    • Figures 1d, 2a and 2c are missing housekeeping gene.
    • Visualization of the figures could be improved. Some figures look unproportioned, so I suggest the authors should unify the style and align the figures.

General remarks:

I support publishing paper Sestrin2 expression has regulatory properties and prognostic value in lung cancer after revision, as a report of a so far poorly investigated co-regulation of Sestrin2 and EMT, stem cell, and drug resistance markers even though it does not prove correlation between them.

Author Response

The response letter for Reviewer 1 was attached.

Reviewer 2 Report

This paper by Chae et al. describes the potential roles of Sestrin2 in proliferation and migration of A549 non-small cell lung cancer (NSCLC) cell line. They show that knockdown of sestrin2 suppressed the proliferation, migration, sphere formation, and drug resistance of A549. In addition, they further virtually analyzed the expression and prognosis of Sestrin2 in human lung cancer via digging several public cancer data portals using several bioinformatics approaches and showed that Sestrin2 is overexpressed and positively correlated with the survival of lung cancer patients, which is inconsistent with the conclusion of a previous study (Am J Transl Res. 2016 Apr 15;8(4):1903-9. eCollection 2016) that demonstrated sestrin2 was a favorable prognostic factor in NSCLC patients. The manuscript is generally descriptive; it doesn't offer essential insight into the mechanisms and significance of sestrin2 in NSCLC. The novelty and significance of the study are limited.

Major concerns
1. All experiments were done on the A549 cancer cell line only, and some key experiments may be repeated in another cancer cell to confirm the findings are biologically reproducible.
2. Changes of genes’ expression at both mRNA and protein levels should be shown in cancer cells.
3. The impact of sestrin 2 on the in vivo growth of NSCLC should be investigated.
4. The underlying mechanisms of sestrin 2 especially these key cancer signaling pathways that are regulated by knockdown of sestrin2 should be intensively explored.
5. Data on genetic alterations of sestrin 2 is not tightly relevant to the study since the correlation and significance of these alterations aren’t known and discussed.

Author Response

The response letter for Reviewer 2 was attached~

Round 2

Reviewer 1 Report

In the revision, authors provide more details regarding experimental conditions for reproduciblity. However, several important suggestions, such as ROS measurement under various conditions, comparison of protein expression in wt and KD cells, have been ignored by authors. I also could not fine the result related rescue effect by re-expressing SESN2.

The reason why i asked for more experimental result instead of bioinformatic analysis in this case, because in most cases the results obtained from bioinformatics are a mixture of various experiments under various experimental conditions. For instance, the role of SESN2 in Lunger cancer are differently using different gene expression data (http://dna00.bio.kyutech.ac.jp/PrognoScan/). Indeed, just 2 out of 16 are statistic significant.

Author Response

A word file was attached~
